# Postnatal Migration of Cerebellar Interneurons

**DOI:** 10.3390/brainsci7060062

**Published:** 2017-06-06

**Authors:** Ludovic Galas, Magalie Bénard, Alexis Lebon, Yutaro Komuro, Damien Schapman, Hubert Vaudry, David Vaudry, Hitoshi Komuro

**Affiliations:** 1Normandie University, UNIROUEN, INSERM, Regional Cell Imaging Platform of Normandy (PRIMACEN), 76000 Rouen, France; magalie.benard@univ-rouen.fr (M.B.); alexis.lebon@univ-rouen.fr (A.L.); damien.schapman@univ-rouen.fr (D.S.); hubert.vaudry@univ-rouen.fr (H.V.); david.vaudry@univ-rouen.fr (D.V.); 2Department of Neurophysiology, Donders Centre for Neuroscience, Radboud University, Nijmegen 6525 AJ, The Netherlands; y.komuro@neurophysiology.nl; 3Department of Neuroscience, School of Medicine, Yale University, New Haven, CT 06510, USA; hitoshi.komuro@yale.edu

**Keywords:** cerebellum, postnatal development, migration, interneuron, live-cell imaging, environmental conditions, drug of abuse, cerebellar disorders

## Abstract

Due to its continuing development after birth, the cerebellum represents a unique model for studying the postnatal orchestration of interneuron migration. The combination of fluorescent labeling and ex/in vivo imaging revealed a cellular highway network within cerebellar cortical layers (the external granular layer, the molecular layer, the Purkinje cell layer, and the internal granular layer). During the first two postnatal weeks, saltatory movements, transient stop phases, cell-cell interaction/contact, and degradation of the extracellular matrix mark out the route of cerebellar interneurons, notably granule cells and basket/stellate cells, to their final location. In addition, cortical-layer specific regulatory factors such as neuropeptides (pituitary adenylate cyclase-activating polypeptide (PACAP), somatostatin) or proteins (tissue-type plasminogen activator (tPA), insulin growth factor-1 (IGF-1)) have been shown to inhibit or stimulate the migratory process of interneurons. These factors show further complexity because somatostatin, PACAP, or tPA have opposite or no effect on interneuron migration depending on which layer or cell type they act upon. External factors originating from environmental conditions (light stimuli, pollutants), nutrients or drug of abuse (alcohol) also alter normal cell migration, leading to cerebellar disorders.

## 1. Introduction

Over the last 100 years, the cerebellum has served as a model system to elucidate the cellular and molecular mechanisms underlying the development of the entire brain. In particular, several seminal studies initially used the developing cerebellum as a suitable model to elucidate the fundamental principles of neuronal migration. The concept of neuronal migration during brain development was first proposed for cerebellar granule cells. In the early 1900s, based on a study using developing cerebella with Golgi staining, Ramon y Cajal assumed that granule cells translocate from their birth place to their final destination, where they reside during their entire adult life, implying that the migration of immature neurons is essential for the formation of normal neuronal cytoarchitecture [1]. More than half a century later, the proof of the existence of neuronal migration in the developing brain was provided for the first time by a study using the developing cerebellum. In 1961, using developing mouse cerebella and tritiated thymidine, Miale and Sidman experimentally revealed the precise time-course of cerebellar granule cell migration [2]. Ten years later, the first discovery regarding the cellular mechanisms underlying neuronal migration was made from a study examining the migration of cerebellar granule cells. Using developing monkey cerebella and electron microscopy, Rakic established that migrating granule cells use the processes of Bergmann glia as a scaffold, leading to the radial glia hypothesis for neuronal migration [3]. Thereafter, many studies followed to search for the molecules which play a role in neuron–glia interaction. Among the studies, using cultured cerebellar neurons and an in vitro assay system for cell movement, Edmondson and Hatten revealed for the first time a critical role of cell adhesion molecules (astrotactin) in the glia-associated migration of immature neurons [4]. It has been subsequently demonstrated that the cellular and molecular mechanisms underlying the control of neuronal migration in the developing cerebellum can be generalized to the migration of many neurons in other brain regions, including the developing cerebrum, with few differences. The gathered knowledge illustrating the precise mechanisms of cerebellar development will in turn contribute to the development of therapeutic strategies aimed at curing brain disorders.

## 2. Postnatal Cerebellum

The cerebellum is a complex foliated structure with ten lobules and a species-specific number of sublobules (Figure 1A) [5]. During development, the cerebellar cortex is organized in four layers including the external granular layer (EGL), the molecular layer (ML), the Purkinje cell layer (PCL) and the internal granular layer (IGL) (Figure 1B) [6]. By the time of birth, the γ-aminobutyric acid (GABA)ergic Purkinje cells which are the only projection neurons in the cerebellar cortex occupy their final position in the PCL between the EGL and the IGL. However, monolayer arrangement of Purkinje cells is only completed at postnatal days 4–5 (P4–P5) in mice [7]. By the third postnatal week, the EGL disappears and the IGL becomes the granular layer (GL) after the completion of cerebellar development [8,9].

Six different types of interneurons originating from two primary germinal zones—the upper rhombic lip (uRL) and the cerebellar plate ventricular zone (VZ)—are produced sequentially (Figure 2) [10]. Most interneuron precursors migrate out the uRL or the VZ to a secondary germinative zone i.e., the EGL, the prospective white matter (pWM) or the deep white matter (dWM) [10].

From these zones, they move to their final destination within the ML or in the IGL during the first three-postnatal weeks (Figure 1B and Figure 2). Two types of excitatory/glutamatergic interneurons—the granule cells (G) and the unipolar brush cells (U)—originate from the uRL. While granule cell precursors move to the EGL, unipolar brush cells migrate directly to the IGL through the white matter (Figure 1B). The majority of unipolar brush cells reach the IGL by P10 in mice [11]. In contrast, the last granule cells reach their final position in the IGL around P20 in mice (Figure 2) [11]. Four inhibitory/GABAergic interneurons—the stellate cells (S), the basket cells (B), the Golgi cells (Gi) and the Lugaro cells (L)—originate from the VZ (Figure 2) [11]. Golgi cells migrate from the dWM through the folial white matter to the IGL until P4 in mice. Basket and stellate cells migrate from the dWM through the folial white matter, the IGL and the PCL to their final destination in the ML until P16 in mice [9]. Lugaro cells migrate through the folial white matter to their final location at the top of the IGL by P5 [12]. The cerebellar cortex therefore represents a unique model for studying postnatal migration with centripetal migration of granule cells and centrifugal migration of unipolar brush cells, basket and stellate cells, Golgi cells, and Lugaro cells. During the first three postnatal weeks, the most numerous migrating interneurons are granule cells and basket/stellate cells with respective densities of 1124 ± 138 and 306 ± 79 cell/mm^2^ in rat P10 cerebellar slices [13]. In this review, we will describe, as a model system of neurodevelopment, the concomitant migration of granule and basket/stellate cells that have distinct embryological origins, opposite migration directions, and opposite effects on Purkinje cell activities during adulthood [14]. In particular, we will focus on ex vivo/in vivo real-time observations of the highway network of migration with saltatory movements, transient arrests or standby phases of interneurons. Then, we will review the multifactorial control of interneuron movements including the role of external factors and the putative disorders provoked by an alteration of the migration.

## 3. Complementary Imaging Approaches for Cell Migration Studies in the Postnatal Cerebellum

Interneuron migration has been studied during the last 25 years through fluorescent micro- or macroscopy for ex vivo or in vivo experiments [15]. Membrane or cytoplasm staining as well as fluorescent protein expression strategies were developed to track cell movements [15]. In particular, lipophilic membrane tracers such as long-chain dialkylcarbocyanines including 1,1′-dioctadecyl-3,3,3′3′-tetramethylindocarbocyanine perchlorate (DiI) were originally used in cerebellar slices to monitor granule cell migration in fine detail [16]. Then, cytosolic probes of the Cell Tracker™ family and cell-expressed fluorescent proteins increased the number of labelled cells and consequently the number of tracked granule cells or basket/stellate cells [13,17,18]. For their simplicity of manipulation, exogenous fluorescent probes were usually used for both ex vivo and in vivo studies [15,19,20] while monocolour genetic labelling of interneurons was only applied for tissue slices [18]. Contrary, the combination of several Cell Trackers™ or Brainbow transgenes [21,22] has never been used for multiplex labelling and may represent a new perspective to study cerebellar interneuron migration.

Several imaging systems including inverted widefield/confocal microscopy as well as upright confocal macroscopy have been used to study interneuron migration in tissue slices (Figure 3). In contrast to widefield microscopy, an increased signal-to-noise ratio and flexibility of dye excitation and detection are the main advantages of confocal approaches (Figure 3). Nowadays, supercontinuum lasers have been shown to provide a flexible excitation source for fluorescence imaging [23], and preservation of the biological sample/fluorochrome can also be improved by decreasing the laser power and by increasing the detection of emitted photons. A combination of vacuum-tube and semi-conductor technologies known as hybrid detectors has recently emerged and outperforms other sensors in most applications [24]. A recent alternative and original approach is confocal macroscopy that offers a large field of view and long working distance (Figure 3). Interestingly, an important number of labelled interneurons can be simultaneously observed in a cerebellar slice placed on the membrane of a Transwell™ insert [15]. The thickness of tissue slices (~200 µm) is fully consistent with the z-depth imaging of confocal approaches. In addition, several types of in vivo tracking of cerebellar interneurons have been successfully performed with inverted confocal microscopy (Figure 3) [20,25]. However, two-photon fluorescence microscopy may also offer several advantages over conventional imaging techniques since it facilitates deep tissue penetration and dramatically reduces the photobleaching of fluorophores (Figure 3) [26]. Although two-photon fluorescence microscopy has been performed on adult mouse cerebellum to investigate axon regeneration [27], it has never been used to study cerebellar interneuron migration in postnatal animals. Light-sheet microscopy is now considered an emerging imaging technology in the field of neurosciences with large scale neuronal recordings. Since transparency of the tissue is mandatory for successful imaging, a small but important number of studies have been performed in living tissue or animal models including larval zebrafish [28,29]. Recently, the zebrafish has emerged as a useful model organism for cerebellar studies, owing in part to the similarity of the cerebellar circuits in between zebrafish and mammals, and Upstream Activator Sequence (UAS) constructs have been used to generate fluorescent cerebellar cells [30]. In the future, adaptation of light-sheet microscopy combined with genetic manipulation of organisms to make tissues more transparent would allow new perspectives to study larger rodent brains [29].

## 4. Cellular Highway Networks within the First Two Postnatal Weeks of the Cerebellum

From multiple previous studies, it is now clear that granule cells and basket/stellate cells exhibit respectively centripetal and centrifugal migration over the same developmental period and in particular during the first two postnatal weeks [6,14].

With four major steps of migration, granule cells pass through all the cortical layers of the cerebellum. Firstly, granule cells migrate tangentially within the EGL and then change direction to migrate radially along the processes of Bergmann glial cells through the ML [31,32] (Figure 4A, blue). When entering the PCL, granule cells detach from glial cells and slow down [31]. Two hours later, granule cells resume their migration and cross the border between the PCL and the IGL. Within the IGL, granule cells migrate radially, independently of glial fibres, until they reach their final position at the bottom of the IGL [31] (Figure 4A, blue).

Basket cells and stellate cells exhibit a centrifugal move during early postnatal development from the pWM to the ML [9,33]. During their migration in the ML, basket cells and stellate cells are generally identified as homogeneous immature interneurons because they are morphologically indistinguishable [18]. However, we cannot rule out the possibility that basket cells and stellate cells exhibit discrete differences in their speed and regulation of migration. In mice [18] and in rats [13], a large majority (80–90%) of basket/stellate cells display a radial migration, whereas only 10–20% of the cells migrate obliquely toward the top of the ML. Basket/stellate cells that display radial migration complete their migration in four phases [9,19,34,35,36]. During phase I, the cells migrate radially from the bottom to the top of the ML where they sojourn next to the EGL [37] (Figure 4A, pink). Then, the cells turn at the top of the ML and migrate tangentially (phase II) with occasional reversal of the direction of migration. At phase III, the cells turn and migrate radially within the ML at reduced speed. Finally, the cells turn at the middle of the ML and migrate tangentially at their slowest speed during the terminal phase IV [18].

Besides the characterization of each cell type profile, our understanding of the general orchestration of interneuron migration is very rudimentary and represents a challenging task. In particular, granule cells and basket/stellate cells have to follow long and complex migratory highways before reaching their final destination. Paradoxically, the starting point of granule cells in the EGL is very close to the ending point of basket/stellate cells in the ML, and, reciprocally, the starting point of basket/stellate cells in the pWM is also very close to the ending point of granule cells in the IGL (Figure 1). The prolonged migration of interneurons may provide a time for cells to form processes and networks in the different cerebellar cortical layers, for instance the formation of parallel fibers for granule cells [11]. Due to the opposite migration routes of granule cells and basket/stellate cells in the ML during the second postnatal week, simultaneous tracking was possible with the use of only a single fluorescent marker [13]. Therefore, simultaneous observation of granule cells and basket/stellate cells through confocal macroscopy is a starting point to study interneuron migratory highways (Figure 4A).

As a new concept in cerebellar interneuron migration, we have shown that basket and stellate cells intersect with granule cells, suggesting that they may share a common migratory rail [13]. Several studies have shown that granule cells migrate radially along the processes of Bergmann glial cells through the ML [38] but this is only speculative for basket/stellate cells, and future studies are required to clarify this point. Regarding basket/stellate cells, that migrate obliquely toward the top of the ML [13,18], it is very unlikely that they move along the processes of glial cells. Taken together, the data suggest that some basket/stellate cells may use different tracks with proper mechanisms and specific regulations to attain their final position in the ML. A potential novel research field is the possible existence of waves of interneuron migration i.e., coordinated movements of groups of granule cells within a lobule or between adjacent lobules. Whether the migration of the different types of interneurons is coordinated and possibly orchestrated by “band leaders” such as Purkinje cells may also be a fruitful hypothesis. Large field of view over long time-lapses used alongside future technological and labelling strategies, previously discussed in this review, will open new possibilities in the topic.

At a cellular level, the mechanisms of granule cell migration have been extensively studied and the saltatory movements resulting from the repetitive translocation of the nucleus and the organelles along the leading processes are clearly established in the EGL, the ML and the IGL [31,32,39] (Figure 4B). Similarly, basket/stellate cells reach the top of the ML with saltatory movements as described in mouse and rat postnatal cerebellar slices [13,18] (Figure 4C). Another consideration is the cell-cell contacts, and possible communications, that cerebellar interneurons might establish with congeneric cells or with other types of interneurons, neurons or glial cells during their long migratory route. The most described interaction is the contact of granule cells with Bergmann glial fibres during the migration in the ML [38]. In addition, along this cellular highway, granule cells also intersect with basket/stellate cells and some of them tend to slow down after crossing [4], suggesting that their interaction may affect granule cell migration. Interaction of granule cells and basket/stellate cells with Purkinje cell soma in the PCL has been poorly characterized so far, although the crossing of the PCL, in an opposite direction, is an obligatory step for both interneurons [6,18]. In fact, the situation is different for granule cells and basket/stellate cells. For example, when entering the PCL, granule cells detach from glial cells, slow down and stop their migration for an approximately two-hour period of time [31]. In contrast, basket/stellate cells coming from the IGL directly cross the PCL without any transient arrest phase [18]. Super-resolution approaches including STimulated Emission Depletion (STED) microscopy [40,41,42] and correlative microscopy [43] could now offer new possibilities to decipher the roles of cell–cell contacts between granule cells, basket/stellate cells, Purkinje cells, and Bergmann glial cells in key steps of cerebellar interneuron migration including the crossing of the PCL.

## 5. Transient Arrest Phase, Standby Phase and Final Stop Signal for Cerebellar Interneuron Migration

Transient arrest phases of various duration appear to be a frequent feature of neuronal migration in brain histogenesis [44]. During postnatal development of the cerebellum, the presence of pauses during interneuron migration is well established for granule cells and basket/stellate cells but their functional significance remains largely unknown.

Once the granule cell soma enters the PCL, its shape transforms abruptly from a vertically elongated spindle to a sphere [6]. These rounded somata slow their movement significantly, and stop completely in the PCL [6]. The transient stationary phase in the PCL is caused, at least in part, by the release of endogenous pituitary adenylate cyclase-activating polypeptide (PACAP) expressed in the soma and the dendrites of Purkinje cells as observed in mice and rats [13,17] (Figure 5).

After a prolonged stationary period with times ranging from 30 to 220 min, the granule cells in the PCL begin to re-extend their somata and leading processes [6]. Furthermore, during this transformation, granule cells gradually accelerate the rate of their migration and cross the border between the PCL and the IGL through a PACAP-induced tissue-type plasminogen activator (tPA) release that degrades the extracellular matrix [13]. The transient arrest phase of granule cells in the PCL could be necessary for initiation of a differentiation program and/or for a correct integration of granule cells in the IGL. In particular, the somatostatinergic system exerts a stimulatory effect on tangential granule cell migration in the EGL but somatostatin is the stop signal of granule cells in the IGL [45]. Therefore, PACAP could be responsible for the switch of the somatostatinergic control of granule cell migration.

At the end of the centrifugal radial migration, basket/stellate cells are still immature and stay next to the EGL for more than 16 hours (phase II) with fluctuations of cell movement including slow speed (2.4 µm/h) or reversal of direction. This “standby phase” of interneurons at the top of the ML might be regulated by netrin1 that is highly expressed in the EGL and that exerts a repulsive action on basket/stellate cells [46] (Figure 5). The meaning of the “standby phase” is not clearly determined but again the initiation of a differentiation program may be required for proper settling of basket/stellate cells within the ML. Another possibility is that basket/stellate cells remain temporarily at the top of the ML during the final establishment of parallel fibres of granule cells to which they will be ultimately connected [34]. After phase II, basket/stellate cells translocate through complex migratory routes to their final destination in the middle part of the ML [18,37]. In contrast to granule cells, there is no clear evidence for stop signal(s) of basket/stellate cell migration. The presence of PACAP in Purkinje cell dendrites and its differentiating effect on stellate morphology of astrocytes [47] suggest that this neuropeptide might be a potential candidate signal for the completion of basket/stellate cell migration.

## 6. Cortical-Layer-Specific Effects of Regulatory Factors during Interneuron Migration

A number of regulatory factors including neurotransmitters (glutamate), neuropeptides (somatostatin, PACAP), extracellular matrix-related proteins (matrix metalloproteinase-3 (MMP-3); tissue plasminogen activator (tPA); tenascin), glycoproteins (astrotactin), growth factors (brain-derived nerve factor (BDNF); neurotrophin-3 (NT-3); neuregulin; insulin growth factor-1 (IGF-1)), cytokines (stromal cell-derived factor-1α (SDF-1α)), glycosphingolipids (9-*O*-acetyl ganglioside3, 9-*O*-acetyl GD3) and phospholipids (platelet-activating factor (PAF)) have been identified over the last 25 years as they facilitate (in blue) or exert a direct stimulatory activity (in green) or a direct inhibitory effect (in red) on the migration of interneurons in different cortical layers of the postnatal cerebellum (Figure 6). Diverse and complementary methods have been used to investigate the effects of such factors by: (1) the inhibition of in situ expressed molecules or receptors by exogenous inhibitors, blockers or antagonists; (2) the silencing of genes in knock-out animals; and (3) the administration of exogenous synthetic or purified compounds with potential effect. In particular, the role of somatostatin [45], PACAP [13,17], tPA [13] and IGF-1 [25] on interneuron migration has been examined ex vivo or in vivo by real-time monitoring of the cell after inhibition of the endogenous molecule (Figure 6). Mainly by comparing the thickness of cerebellar cortical layers, the effects of MMP-2/3/9 [48,49,50], BDNF [51], SDF-1 [52], tPA [53] or astrotactin [54] have been evidenced in knock-out (KO) animals or after in vivo patch implantation for neurotrophin-3 [55] (Figure 6). The involvement of netrin-1 [46], tenascin [56], and 9-*O*-acetyl GD3 [57] in the regulation of interneuron movements has been proposed from cerebellar microexplants studies while the role of PAF [58] and neuregulin [59] was observed on cultured cells (Figure 6). Each strategy has limitations that need to be considered including bio-accessibility, toxicity, emerging regulation in KO animals or over-concentration (endogenous plus exogenous) of compounds, and receptor desensitization. Nevertheless, it appears that interneuron migration during postnatal development of the cerebellum is under a multi-factorial control indicating a complex orchestration with cortical-layer-specific effects of regulatory factors.

During postnatal development, things are not set in stone and the roles of regulatory factors depend not only on when and where they act but also on which type of targeted interneuron and through which subtype of receptor they induce intracellular responses. In this respect, the roles of somatostatin and PACAP are characteristic examples of the cortical-layer-specific effect of regulatory factors. Somatostatin is a neuropeptide with two bioactive forms, somatostatin-14 (SST-14) and somatostatin-28 (SST-28) [60,61]. Five somatostatin receptors (SSTRs) have been cloned and named SSTR1 to SSTR5 according to their order of identification [61]. Both SST-14 and SST-28 bind to all five SSTRs. During postnatal cerebellar development, SST-14 is present in Purkinje cells (ML and PCL), Golgi cells, and climbing fibres (IGL), while SST-28 is detected in Golgi cells and mossy fiber terminals (IGL) [45]. Three binding sites, including SSTR1, SSTR4, and mainly SSTR2 are expressed during cerebellar development [62]. *Sstr1* mRNA is predominant at the end of gestation [62]. A transient high expression of the *sstr2* gene is observed from P7 to P14 in the rat whereas *sstr4* mRNA levels are generally low [62]. A SSTR2/SSTR5 antagonist (AC-178,335) significantly decreases the rate of granule cell migration in the EGL, slightly increases the rate in the ML and significantly increases the rate in the IGL [45]. Therefore, somatostatin accelerates the tangential movement of granule cells near the birthplace within the EGL, but significantly slows down radial movement and, in particular, acts as a stop signal within the IGL (Figure 6). These data suggest firstly a diffusion of somatostatin from its source possibly from the dendrites of Purkinje cells in the ML to reach granule cells in the EGL and secondly a switch of the somatostatin receptor/signalling system during the migration of granule cells to become a stop signal inducer in the IGL.

PACAP exists in two bioactive forms: PACAP38 and PACAP27 [63,64,65]. PACAP27 corresponds to the N-terminal 27-amino acid sequence of PACAP38 [63,64]. In the postnatal cerebellum of rodents, PACAP27/38 is expressed sporadically at the bottom of the ML in the dendrites of Purkinje cells, intensively in the somata of the Purkinje cells in the PCL, and throughout the IGL, most likely in the mossy fibre terminals [17,66,67]. Three PACAP receptors have been cloned, and termed pituitary adenylate cyclase activating polypeptide receptor 1 (PAC1), vasoactive intestinal peptide receptor 1 (VPAC1) and vasoactive intestinal peptide receptor 2 (VPAC2) [65]. In the early postnatal rat cerebellum, the expression levels of PAC1 receptors are two to three times higher than those of the VPAC1 receptors, and no VPAC2 receptors can be detected [68,69]. In the EGL, the density of PAC1 receptors is high from birth to P12, and markedly decreases from P12 to P25. In the ML and IGL, PAC1 receptors are first detected at P8. In the ML the density of PAC1 receptors rapidly decreases during the second and third postnatal weeks, and virtually disappears after P25. In the IGL the density of PAC1 receptors slightly decreases during the second and third postnatal weeks. VPAC1 receptors are only expressed at low level in the EGL during the first and second postnatal weeks of the rat cerebellum [69]. A PACAP receptor antagonist (PACAP6-38) accelerates granule cell migration in the PCL, but does not change their migration rate in the EGL, ML and IGL [13,17]. Therefore, despite the wide distribution of PACAP in the ML, the PCL and the IGL, and PACAP receptors in all cortical layers of the cerebellum, the inhibitory effect of PACAP on granule cells migration is restricted to the PCL (Figure 6).

Additional regulatory peptides controlling interneuron migration are likely to be discovered in the near future. For instance, the spatio-temporal expression of preproenkephalin [70] and preprogalanin [71] RNA in Purkinje cells of certain lobules during the first three postnatal weeks offers new perspectives in the understanding of differential development of the anterior and posterior cerebellar lobes.

Radial or tangential cell migration is systematically associated with the degradation of the extracellular matrix (EM) allowing interneurons to move within the different cerebellar cortical layers and to reach their final location. Several components of proteolytic cascades have been identified to play a role mainly in the migration of granule cells. tPA is an extracellular serine protease that converts the proenzyme plasminogen into the active protease plasmin, which in turn degrades EM components such as cell adhesion molecules or laminin [72,73]. In situ hybridization and immunohistochemical studies have revealed the presence of tPA mRNA and tPA-like immunoreactivity in the ML, the PCL, the IGL, and the white matter (WM) of the postnatal cerebellum [13,74,75]. In contrast, the EGL is virtually devoid of immunoreactive signals. In particular, tPA is detected in leading processes of migrating granule cells [76,77,78,79,80]. As a matter of fact, plasminogen mRNA is widely expressed in the cerebellar EGL, ML and IGL of newborn mice [79]. Time-lapse macroconfocal imaging indicates that plasminogen activator inhibitor-1 (PAI-1), an inhibitor of tPA, reduces the velocity of granule cell by 70% in the ML and by 27% in the PCL but has no effect in the IGL. The developing cerebellum of tPA-deficient mice also exhibits an increased number of granule cells in transit within the ML as a result of a decrease in migration speed [53]. In the ML, two types of basket/stellate cells have been identified according to their radial migration speed and were designated as slow (68%) and fast (32%) cells [13]. In the presence of PAI-1, 94% of basket/stellate cells exhibit a slow migration speed profile, indicating that endogenous tPA participates to the radial centrifugal migration of fast cells, whereas the migration of slow cells is not dependent on tPA. Therefore, endogenous tPA contributes to the glia-dependent radial migration of granule cells in the ML and to the glia-independent radial migration of granule cells in the PCL. tPA also facilitates centrifugal migration of fast basket/stellate cells in the ML (Figure 6).

Among the matrix metalloproteinase (MMP) family, MMP-2, MMP-3 and MMP-9 are detected in the postnatal cerebellum and appear to play a role in the migration of granule cells by degrading the EM [48,49,50]. Despite a wide distribution of MMP mRNAs or immunoreactive materials during the postnatal migratory phase of interneurons, these proteinases seem to regulate only the tangential migration of granule cells in the EGL [48,49,50]. (Figure 6). However, MMP-3 may act through different mechanisms in the EGL by degrading tenascin, but also SDF-1α and BDNF [50]. Therefore, the role of MMPs appears to be restricted to the EGL by facilitating granule cells to reach the ML.

## 7. Impact of Environmental Conditions, Pollutants, Nutrients, and Drug of Abuse on Interneuron Migration 

In humans, the development of the cerebellum starts around the fourth week of the embryonic phase with the formation of the cerebellar primordium and lasts up to the first postnatal year [80]. By sharing both embryonic and postnatal phases of development, the cerebellum is therefore a very suitable model to study interneuron migration in animals that can be extrapolated to humans. During gestation or lactation, the cerebellum is susceptible to environmental stimuli (light, sound, temperature) [81,82,83], toxic compounds (MeHg, Pb) [84,85,86,87], nutritional deficiency (Zn, Cu) [88], drug consumption (alcohol, cocaine, amphetamine/methamphetamine, nicotine) [89], or medications (antidepressants, antipsychotics) [89], that may affect particularly the migration of cerebellar interneurons.

As previously described, the majority of cerebellar granule cells migrate after birth in both humans and mice [6,90]. Even though mice do not open their eyes until approximatively P12, it has been reported that light stimuli induce physiological changes before that time [91]. In vivo experiments indicate that tangential migration speed of granule cells in the EGL increases during the light phase and decreases during dark phase. In addition, cerebellar levels of IGF-1 are high during light phases and low during dark phases [20]. Furthermore, inhibition of IGF-1 receptors by picropodophyllin, a specific IGF-1 receptor inhibitor, during the light phase, decreases the speed of granule cell migration. In contrast, administration of exogenous IGF-1 during the dark phase, increases the rate of granule cell migration. Consequently, there are causal relationships between light-dark cycles, cerebellar IGF-1 levels and speed of granule cell migration [20]. Obviously, complementary studies are needed to determine whether similar effects of “light-stimulation” on neuronal migration could be applied to preterm and full-term human babies, but this may change and standardize the current approach of light exposure of infants at hospitals and homes.

Mercury-containing industrial wastes dumped in rivers and seas could result in the accumulation of methylmercury (MeHg) in fish. Consumption of MeHg-contaminated fish by women transfers MeHg to the foetus through the placenta or to developing infants via breastfeeding. The foetus behaves as a “mercury trap” and the concentration of MeHg in the fetal brain is at least twice that of the mother’s [92]. Known as foetal Minamata disease (FMD), brains of patients that have been exposed to MeHg during development present notably, but not exclusively, neuropathological characteristics in the cerebellum [93,94,95]. Ex vivo experiments have revealed that MeHg (20 µM) inhibits the migration of granule cells in the EGL, ML and IGL in a dose-dependent manner. MeHg decreases the speed of granule cell migration by reducing the occurrence of spontaneous Ca^2+^ spikes and stimulating the 3′,5′-cyclic adenosine monophosphate (cAMP) pathway [20]. Manipulating Ca^2+^ and/or cAMP signaling pathways by application of caffeine, Rp-Cyclic 3′,5′-hydrogen phosphorothioate adenosine triethylammonium salt (Rp-cAMPS) (a competitive cAMP antagonist) or IGF-1 significantly reduces the effects of MeHg on granule cell migration, suggesting a potential therapeutic strategy for infants with MeHg intoxication. Furthermore, it has been reported that, during brain development, lead (Pb) intoxication through urban pollution can also induce alterations of neurobehavioral outcomes under cerebellar control although the cellular mechanisms involved remain to be elucidated [88].

A wide variety of nutrients play an important role in neuronal development, but some of them, including proteins, iron, zinc, selenium, iodine, folate, vitamin A, choline and long-chain polyunsaturated fatty acids, appear to have greater effects during the late foetal and early postnatal life [88]. The cerebellum is particularly vulnerable to early postnatal undernutrition [96], although the underlying cellular mechanisms remain largely unknown. Zinc and copper deficiency appear to be a risk factor for the developing cerebellum with long-term effects on motor function, balance and coordination [97,98]. Although, to date, the question of whether zinc and copper affect interneuron migration remains to be examined, there are interesting clues. Indeed, zinc regulates the expression of the *igf-1* gene [99], and a positive correlation exists between the expression of IGF-1 and cerebellar grey matter volume and mental development [100]. Collectively, these observations suggest that zinc deficiency affects interneuron migration maybe by reducing the IGF-1 levels, leading to the disrupted development of cerebellar cortical layers.

The effects of prenatal exposure to illegal drugs (cocaine, amphetamine/methamphetamine), legal drugs (nicotine, alcohol), and medications (antidepressant, antipsychotics, relaxing agents) on brain development are complex and dependent on the time, dose and route of drug administration. As far as we know, studies depicting the impact of drugs on the developing cerebellum only concern “foetal alcohol spectrum disorders” (FASDs) and “foetal alcohol syndrome” (FAS) [101]. Prolonged exposure to alcohol during gestation and lactation leads to a pattern of abnormal development in newborns, including cerebellar damage [102,103,104,105,106]. Alcohol exposure results in abnormal development of the postnatal cerebellum [107,108,109]. On cerebellar slices, ethanol reduces the speed of granule cell migration in the EGL, ML and IGL in a dose-dependent manner [110]. Furthermore, intraperitoneal injection of ethanol affects the direction of granule cell migration from tangential to radial at the EGL–ML border [111]. Ethanol reduces the frequency of spontaneous Ca^2+^ transients and 3′,5′-cyclic guanosine monophosphate (cGMP) levels, but increases cAMP levels in migrating granule cells [110,111]. Interestingly, experimental manipulations of Ca^2+^, cGMP and cAMP signalling significantly ameliorate the effects of ethanol on granule cell migration.

## 8. Conclusions

The cerebellum has reciprocal anatomical connections with the cortical motor, frontal, parietal and limbic areas, and is consequently involved in action, cognition, emotion, perception and interoception [80]. Several disorders have their origin in neurodevelopment, and a good understanding of cerebellar development is crucial to unravel the aetiology of different pathologies. Neurodevelopmental defects of the cerebellum have various origins such as stroke (preterm birth haemorrhage), genetic mutations or protein deficiency, pollutants, or drugs that can alter interneuron migration. In addition, there is a wide variety of cerebellum-linked disorders including autism spectrum disorders, neuropsychiatric illnesses (schizophrenia), encephalopathy, hypoplasia, cerebellar ataxia, FMD, FASDs, FAS, stuttering, diabetes or medulloblastoma [80,112,113] and emerging strategies with human induced pluropotent stem cells (hiPSCs) are now proposed for regenerative neurology [114]. How the cerebellum grows and develops, when cell populations have particular vulnerabilities, and how we identify the complications of pathological development, are crucial questions and fascinating research avenues. To date, the cellular and molecular mechanisms underlying the termination of interneuron migration are still largely unknown. There are six types of cerebellar interneurons that migrate during the first three postnatal weeks. Most studies conducted so far, concern granule cells and basket/stellate cells that are quite different by their origin, final destination and role, but share common routes and characteristics of migration and possibly interact. In particular, they both migrate late during the process of development and have the longest ways to go from their own germinative zone to their respective final destination. Interneurons are under multifactorial control with neuropeptides, growth factors, cytokines, glycoproteins, lipids that stimulate or inhibit their migration in a cortical-layer specific manner. In addition, proteolytic cascades degrade the extracellular matrix to facilitate the movement of interneurons. The general orchestration of interneuron migration during postnatal cerebellar development could possibly be driven by the only projection neurons, the Purkinje cells [5], that are first in place around P4 and that express a number of regulatory factors including somatostatin, PACAP, IGF-1, tPA, matrix metalloproteinases (MMPs), encephalin, and galanin. New technological developments to fluorescently label and image the whole (living) cerebellum are certainly needed to achieve, in the future, a full comprehensive understanding of interneuron migration.

## Figures and Tables

**Figure 1 brainsci-07-00062-f001:**
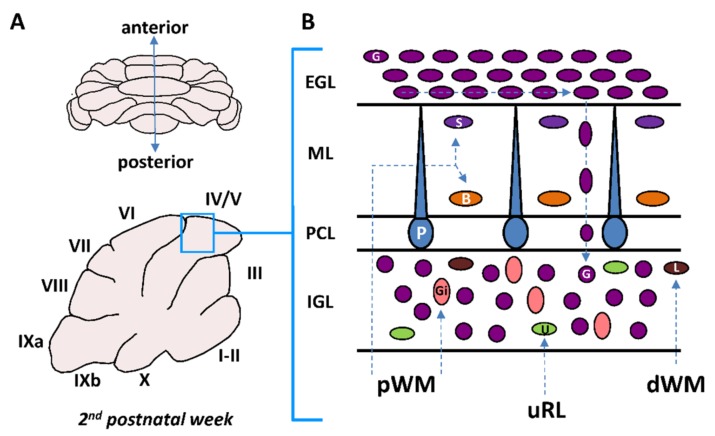
The cortex of the cerebellum: a unique model to study postnatal migration. (**A**) Complex foliated structure of the postnatal cerebellum with lobules (I–X) as shown in schematic sagittal section. IXa and IXb are sublobules of lobule IX. (**B**) Interneuron migration: from germinative zones to final location. dWM, deep white matter; pWM, prospective white matter; uRL, upper rhombic lip; EGL, external granular layer; ML, molecular layer; PCL, Purkinje cell layer; IGL, internal granular layer; Gi, Golgi cell; U, unipolar brush cell; G, granule cell; L, Lugaro cell; P, Purkinje cell; B, basket cell; S, stellate cell.

**Figure 2 brainsci-07-00062-f002:**
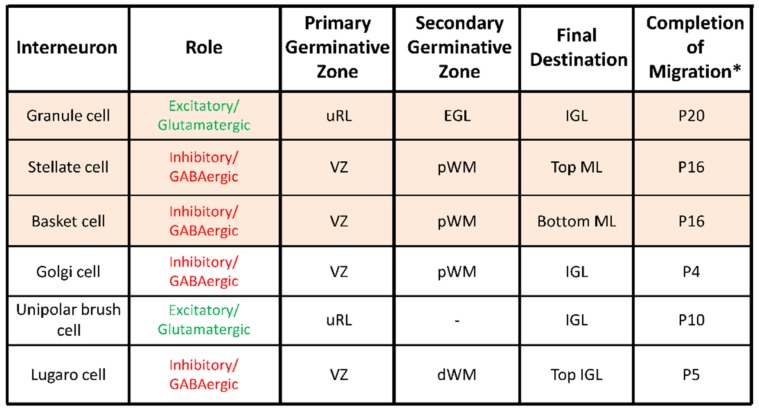
Characteristics of migrating interneurons in the postnatal cerebellum. dWM, deep white matter; pWM, prospective white matter; uRL, upper rhombic lip; EGL, external granular layer; ML, molecular layer; PCL, Purkinje cell layer; IGL, internal granular layer; VZ, ventricular zone; P20, postnatal day 20. * in mouse or rat.

**Figure 3 brainsci-07-00062-f003:**
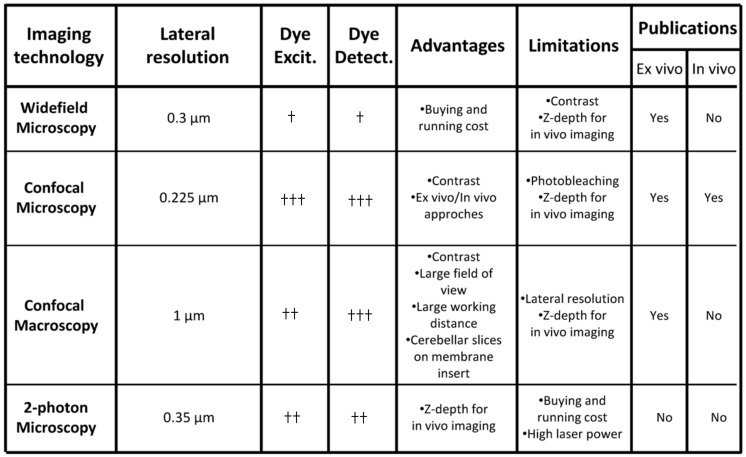
Comparative analysis of imaging technologies used for ex vivo/in vivo interneuron migration studies in the postnatal cerebellum. †, weakly efficient; ††, moderately efficient; †††, highly efficient.

**Figure 4 brainsci-07-00062-f004:**
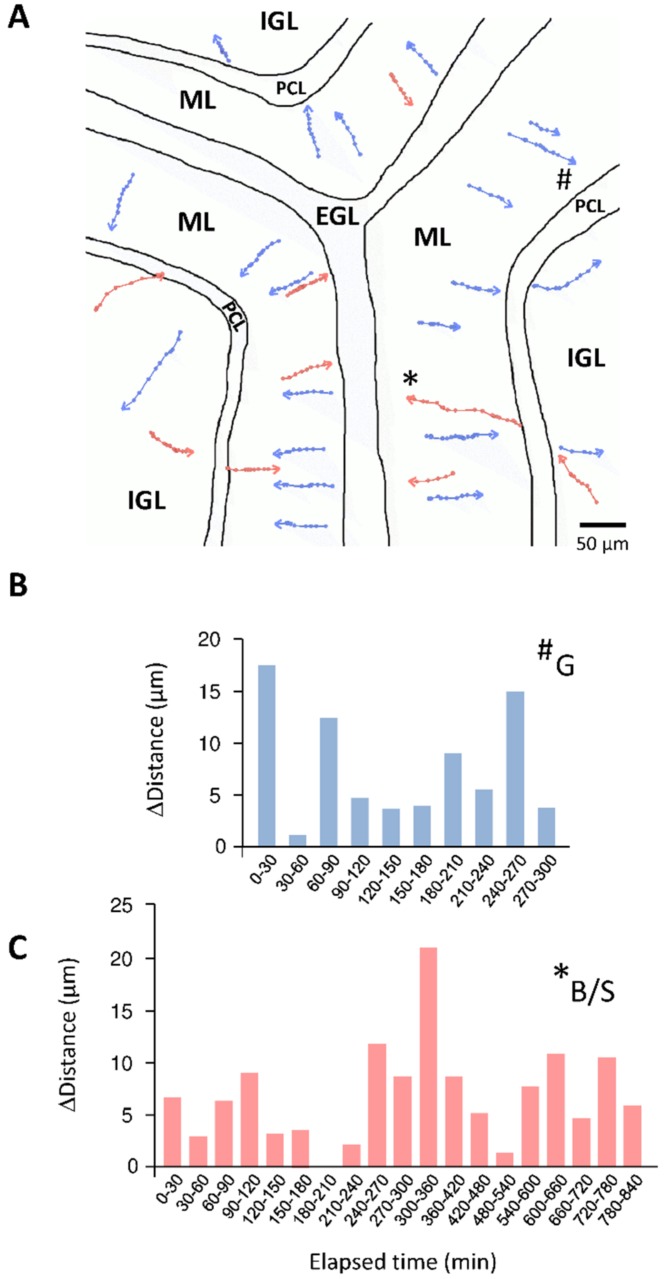
Illustration of the concomitant migration of granule and basket/stellate cells in the postnatal cerebellum through confocal macroscopy. (**A**) Tracking of centripetal migration (towards the bottom of the IGL) of granule cells (blue) and centrifugal migration (towards the top of ML) of basket/stellate cells (pink). (**B**) Saltatory movements of a granule cell (marked as # in (**A**)) in the ML. (**C**) Saltatory movements of a basket/stellate cell (marked as * in (**A**)) in the ML. EGL, external granular layer; ML, molecular layer; PCL, Purkinje cell layer; IGL, internal granular layer.

**Figure 5 brainsci-07-00062-f005:**
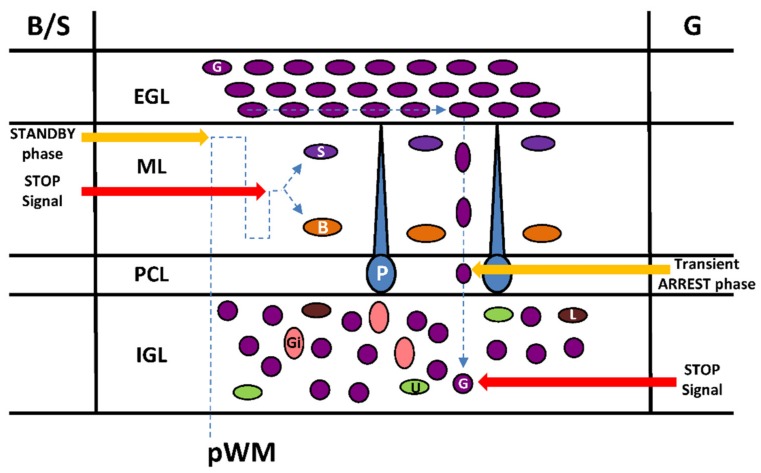
Transient arrest phase, standby phase and stop signals for interneurons during migration in the postnatal cerebellar cortex. pWM, prospective white matter; EGL, external granular layer; ML, molecular layer; PCL, Purkinje cell layer; IGL, internal granular layer; Gi, Golgi cell; U, unipolar brush cell; G, granule cell; L, Lugaro cell; P, Purkinje cell; B, basket cell; S, stellate cell.

**Figure 6 brainsci-07-00062-f006:**
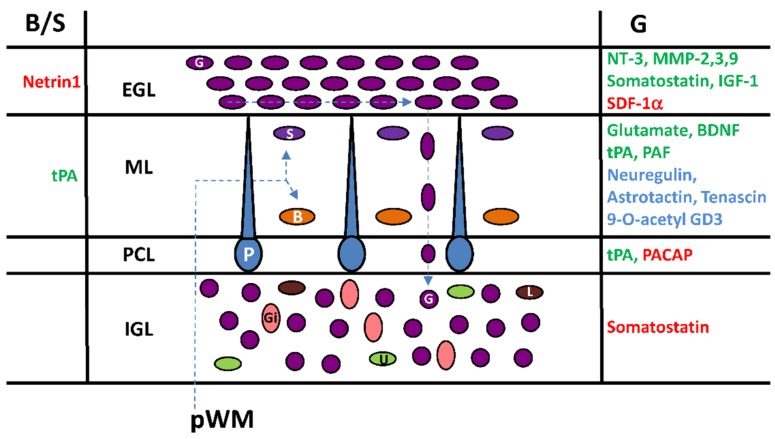
Multifactorial control of interneuron migration in the postnatal cerebellar cortex. Cortical-layer-specific effects of molecules that facilitate (in blue), stimulate (in green) or inhibit (in red) cell movements. pWM, prospective white matter; EGL, external granular layer; ML, molecular layer; PCL, Purkinje cell layer; IGL, internal granular layer; Gi, Golgi cell; U, unipolar brush cell; G, granule cell; L, Lugaro cell; P, Purkinje cell; B, basket cell; S, stellate cell; tPA, tissue-type plasminogen activator; PACAP: pituitary adenylate cyclase-activating polypeptide; NT-3, neurotrophin-3; MMP, matrix metalloproteinase; IGF-1, insulin growth factor-1; SDF-1α, stromal cell-derived factor-1α; BDNF, brain-derived nerve factor; PAF, platelet-activating factor.

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
