# Peer review of "Postnatal Migration of Cerebellar Interneurons"

_brainsci, 2017, doi:10.3390/brainsci7060062_

Round 1

Reviewer 1 Report

In this review the authors discuss the orchestration of cerebellar interneuron migration. They first present the technological means by which migration has been studied in the cerebellum and summarize the key modes of granule and basket/stellate migration and the key factors implicated in this process. They then focus on two or three specific factors as an example of layer targeted control of migration and finally they discuss how interneuron migration can be affected by different exogenous factors during development that cause cerebellar disorders.

Major concern:

The title of this review is promising a more general review on the field of cerebellar interneuron migration that the authors have not managed to convey entirely. Instead there is a continuous focus on specific factors (PACAP, tPA and Somatostatin) which gives the impression that the authors are reviewing the publications from Raoult et al. ,2014 and Yacubova and Komuro, 2002.

More specifically the section on factors affecting interneuron migration (section 5) should be significantly expanded to contain information about more than PACAP, tPA and Somatostatin.

Minor comments:

1.     Line 204: The mode of migration of obliquely oriented basket/stellate cells is not discussed further. Is anything known about it?

2.     Section 3 especially from line 264 to 281 should be rewritten. Some parts make no sense in the current context.

3.     Line 274: ‘…some basket/stellate cells may use different tracks with proper mechanisms and specific regulations to attain their final position.’ What are the proper mechanisms and specific regulations that the authors are referring to here? Some further clarification is required.

4.     Line 313: The authors should also discuss the fact that defective PAKAP signaling in PAKAP receptor mutants leads to increased levels of granule neuron developmental death at their place of origin, the EGL (Morel et al, J. Mol. Neu, 2008). Possibly PACAP expressed by Purkinje cells has a dual or even triple role, initially regulating granule cell survival in the EGL and secondarily boosting migration in the EGL-ML but acting as a STOP signal in the IGL.

5.     Lines 356-358: Do the basket/stellate cells express PACAP receptors?If not it would be rather unlikely that they would use PACAP as a STOP signal like granule cells.

6.     Line 370 ‘’on’’ should be changed to ‘’by’’

7.     Lines 372-380: Instead of listing how each molecule was studied, it would be more interesting to the reader if a small summary of each molecule’s function on granule or basket/stellate cell migration was presented. Figure 4 would then make more sense, as currently the appearance of these molecules next to each cerebellar layer is more indicative of their expression pattern rather than their function. Although the color coding is explained in the text it should also be included in the legend to clarify what the authors want to show in that figure. An expression pattern or a function on that layer?

8.     Neutrin 1 should be corrected to Netrin 1 in Figure 4

9.     Lines 410-446 and 453-474: This entire section is like a review on PACAP and somatostatin. Again it feels the authors are reviewing their latest publication instead of reviewing the field

10.  Line 490: ‘to be exposed’ is redundant in this sentence

11.  Lines 520 -522: This sentence needs to be re-written- ‘also the cellular mechanisms involved remain to be elucidated’ should be moved to the end of the sentence instead and ‘also’ replaced with ‘although’

12.  Lines 533-535: Although this is an interesting hypothesis there is no evidence to support it so the authors might want to include a ‘maybe’ in the sentence.

Author Response

Point-by-point responses

Reviewer: 1

Comments to the Author

In this review the authors discuss the orchestration of cerebellar interneuron migration. They first present the technological means by which migration has been studied in the cerebellum and summarize the key modes of granule and basket/stellate migration and the key factors implicated in this process. They then focus on two or three specific factors as an example of layer targeted control of migration and finally they discuss how interneuron migration can be affected by different exogenous factors during development that cause cerebellar disorders.

Major concern:

The title of this review is promising a more general review on the field of cerebellar interneuron migration that the authors have not managed to convey entirely. Instead there is a continuous focus on specific factors (PACAP, tPA and Somatostatin) which gives the impression that the authors are reviewing the publications from Raoult et al. ,2014 and Yacubova and Komuro, 2002.

More specifically the section on factors affecting interneuron migration (section 5) should be significantly expanded to contain information about more than PACAP, tPA and Somatostatin.

Over the past 20 years, a number of reviews and book chapters have been published in the field of cerebellum development, cell migration, and their regulatory factors. When I received the invitation from brain sciences and from Dr Toyooka as an Academic Editor, my aim was to humbly propose a different review on postnatal migration of cerebellar interneurons with summaries, perspectives and hypotheses. Therefore, partial focus on PACAP, somatostatin and tPA in only 2 sections over 7 in the original version (and now in 2 sections over 8 in the revised version), was necessary to illustrate i) the transient arrest phase, standby phase and final stop signals in section #4 and ii) the cortical-layer-specific effects of regulatory molecules in section #5. In particular, only somatostatin and PACAP have been described as stop signals for granule cells so far. In addition, due to their broad expression in the different cerebellar cortical layers and the detailed studies on their effects on interneuron migration, somatostatin, PACAP and tPA were the best molecules to be considered to illustrate section #5 (in the original version). To avoid any ambiguity as to the focus of this review, that could be caused by the term “orchestration”, we propose to modify the title that now reads “Postnatal migration of cerebellar interneurons”.    

Minor comments:

 1.     Line 204: The mode of migration of obliquely oriented basket/stellate cells is not discussed further. Is anything known about it?

We want to thank the reviewer for this comment. Interestingly, basket/stellate cells that display an oblique migration towards the top of the ML have been observed in mouse (19%) and in rat (10%). Due to the small number of cells, their study is difficult and we actually lack information. These cells are likely to be a subtype of basket or stellate cells but unfortunately we do not know anything about them. In the original submission, from line 271 to line 274, we have suggested that the track, the mechanism, and the regulation of these basket/stellate might be specific.

2.     Section 3 especially from line 264 to 281 should be rewritten. Some parts make no sense in the current context.

Respectfully and from what I understood, one purpose of this Special Issue is to inspire further studies on the mechanisms of neuronal migration, as well as discussion on the methods that will be used for the analysis of neuronal migration. At the beginning of the review process, an abstract and a detailed content have been proposed to the Editorial office including Section 3 entitled “Cellular highway networks within the first two postnatal weeks of the cerebellum” and the strategy received a positive feed-back. As previously mentioned, the aim was to propose a different review allowing readers to consider the simultaneous migration of different interneurons. In addition, we discussed new concepts (common migratory rails, migration of groups of cells, cell-cell contacts) and the necessity to use complementary imaging approaches to have a large field of view or nanoscale resolution. Therefore, we believe that section 3 is fully adapted in the context of this Special Issue on Neuronal migration and Cortical Development.

3.     Line 274: ‘…some basket/stellate cells may use different tracks with proper mechanisms and specific regulations to attain their final position.’ What are the proper mechanisms and specific regulations that the authors are referring to here? Some further clarification is required.

Please see answer to comment #1.

4.     Line 313: The authors should also discuss the fact that defective PAKAP signaling in PAKAP receptor mutants leads to increased levels of granule neuron developmental death at their place of origin, the EGL (Morel et al, J. Mol. Neu, 2008). Possibly PACAP expressed by Purkinje cells has a dual or even triple role, initially regulating granule cell survival in the EGL and secondarily boosting migration in the EGL-ML but acting as a STOP signal in the IGL.

            In this review, we wanted to focus on the migration interneuron. In addition, as previously explained, we did not want to propose a review on specific factors such as PACAP that have already been summarized in other reviews. We do agree with Reviewer # 1 that PACAP may have dual or even triple roles. The fact that one factor can have multiple roles depending on the cortical layer is one of the messages that we wanted to emphasize in this review. However, we did not want to consider other effects (proliferation, differentiation, apoptosis…) that would be out of our scope. Nevertheless, PACAP cannot be considered as a boosting factor in the EGL-ML or as a stop factor in the IGL as previously described (Cameron et al. 2007; Raoult et al., 2014).    

5.     Lines 356-358: Do the basket/stellate cells express PACAP receptors?If not it would be rather unlikely that they would use PACAP as a STOP signal like granule cells.

PACAP usually exerts dual effects on neural/glial cells that is inhibition of migration and stimulation of differentiation. The expression of PACAP receptors in basket/stellate cells has not been studied so far. Consequently, we cannot exclude that PACAP might act as a stop signal for basket or stellate cells in the ML. Lines 356-358, we have only suggested that PACAP might be a stop signal.   

6.     Line 370 ‘’on’’ should be changed to ‘’by’’

Following the reviewer’s request, “on” has been replaced by “by” and the sentence now reads:  Diverse and complementary methods have been used to investigate the effects of such factors by i) the inhibition of in situ expressed molecules or receptors by exogenous inhibitors, blockers or antagonists, ii)…

7.     Lines 372-380: Instead of listing how each molecule was studied, it would be more interesting to the reader if a small summary of each molecule’s function on granule or basket/stellate cell migration was presented. Figure 4 would then make more sense, as currently the appearance of these molecules next to each cerebellar layer is more indicative of their expression pattern rather than their function. Although the color coding is explained in the text it should also be included in the legend to clarify what the authors want to show in that figure. An expression pattern or a function on that layer?

We want to thank the reviewer for pointing out this important point. Again, to focus on migration and to propose a simple and clear message, the role of Figure 4 is only to indicate the function of molecules that act as facilitators (in blue), stimulators (in green) or inhibitors (in red) without considering an expression pattern. To satisfy the reviewer’s request, we have modified the legend to this Figure that now reads: “Figure 4. Multifactorial control of interneuron migration in the postnatal cerebellar cortex. Cortical-layer-specific effects of molecules that facilitate (in blue), stimulate (in green) or inhibit (in red) cell movements.”   

8.     Neutrin 1 should be corrected to Netrin 1 in Figure 4

Neutrin 1 has been replaced by Netrin 1 in Figure 4. We would like to thank the reviewer for pointing out this mistake.

9.     Lines 410-446 and 453-474: This entire section is like a review on PACAP and somatostatin. Again it feels the authors are reviewing their latest publication instead of reviewing the field

 Please see answer to major concern.

10.  Line 490: ‘to be exposed’ is redundant in this sentence

To satisfy the reviewer’s request, we have deleted “to be exposed” line 513 and the sentence now reads: During gestation or lactation, the cerebellum is susceptible to environmental stimuli (light, sound, temperature) [81–83], toxic compounds (MeHg, Pb)…

11.  Lines 520 -522: This sentence needs to be re-written- ‘also the cellular mechanisms involved remain to be elucidated’ should be moved to the end of the sentence instead and ‘also’ replaced with ‘although’

To satisfy the reviewer’s request, we have modified the sentence lines 543-545 that now reads: Furthermore, it has been reported that, during brain development, lead (Pb) intoxication through urban pollution can also induce alterations of neurobehavioral outcomes under cerebellar control although the cellular mechanisms involved remain to be elucidated.

12.  Lines 533-535: Although this is an interesting hypothesis there is no evidence to support it so the authors might want to include a ‘maybe’ in the sentence.

To satisfy the reviewer’s request, ‘maybe’ has been inserted and the sentence lines 556 now reads: Collectively, these observations suggest that zinc deficiency affects interneuron migration maybe by reducing the IGF-1 levels, leading to the disrupted development of cerebellar cortical layers.

Reviewer 2 Report

Postnatal orchestration of cerebellar interneuron migration

In the current manuscript, the authors provide a detailed and accurate description of the neuronal migration in the rodent cerebellum, with special emphasis in postnatal phases. The manuscripts explores initially the technical possibilities given by state-of-the-art microscopy tools, to explore migration in the cerebellum, later focusing in cellular and molecular aspects controlling the distinct types of cerebellar neuronal migration. Finally the authors provide a general insight into how environmental factors  may disrupt these migratory processes and its potential link to disease.

The figures are simple but clear and fully convey the intended message.

Although the manuscript could be enriched by addressing the interesting developments in this area using human-derived models such as iPSC, I consider the manuscript already conveys a thorough and elaborated approach to their intended goal. Moreover, it complies with all the general requirements of the journal  and only requires correction of some minor mistakes, showed here:

266 confocal macroscopy

368 effect (in red) effect

442 receptor antagonist (unintended underlining?)

555 Neurodevelopmental defaults (defects)

Author Response

Reviewer: 2

Comments to the Author

In the current manuscript, the authors provide a detailed and accurate description of the neuronal migration in the rodent cerebellum, with special emphasis in postnatal phases. The manuscripts explores initially the technical possibilities given by state-of-the-art microscopy tools, to explore migration in the cerebellum, later focusing in cellular and molecular aspects controlling the distinct types of cerebellar neuronal migration. Finally the authors provide a general insight into how environmental factors may disrupt these migratory processes and its potential link to disease.

The figures are simple but clear and fully convey the intended message.

Although the manuscript could be enriched by addressing the interesting developments in this area using human-derived models such as iPSC, I consider the manuscript already conveys a thorough and elaborated approach to their intended goal. Moreover, it complies with all the general requirements of the journal and only requires correction of some minor mistakes, showed here:

We want to thank Reviewer #2 for this important comment. The differentiation of human embryonic stem (ES) or induced pluripotent stem (iPS) cells in granule cells or Purkinje cells certainly represents a future technology for regenerative neurology (Wang et al., 2015; Wiethoff et al., 2015). To address this point in the review, we have modified the conclusion (lines 582-583) and the sentence now reads: In addition, there is a wide variety of cerebellum-linked disorders including autism spectrum disorders, neuropsychiatric illnesses (schizophrenia), encephalopathy, hypoplasia, cerebellar ataxia, FMD, FASDs, FAS, stuttering, diabetes or medulloblastoma [80,112,113] and emerging strategies with human induced pluripotent stem cells (hiPSCs) are now proposed for regenerative neurology [114].

266 confocal macroscopy

The right term is actually confocal macroscopy and not confocal microscopy (line 286).

368 effect (in red) effect

We would like to thank the reviewer for pointing out this mistake. The second “effect” has been deleted (line 391).

442 receptor antagonist (unintended underlining?)

We would like to thank the reviewer for pointing out this mistake. Underlining has been deleted (line 464).

555 Neurodevelopmental defaults (defects)

To satisfy the reviewer’s request, “Neurodevelopmental defaults” has been replaced by “Neurodevelopmental defects”(line 577).

Round 2

Reviewer 1 Report

The authors provided satisfactory answers to my concerns and made some necessary changes to the document.